# How Can Apartment-Complex Landscaping Space Improve Residents’ Psychological Well-Being?: The Case of the Capital Region in South Korea

**DOI:** 10.3390/ijerph191610231

**Published:** 2022-08-17

**Authors:** Jongwook Tae, Daeyoung Jeong, Jinhyung Chon

**Affiliations:** 1Division of Environmental Science & Ecological Engineering, Korea University, Seoul 02841, Korea; 2Division of Co-Prosperity & Balanced Development, Gyeonggi Research Institute, Suwon 16202, Korea

**Keywords:** urban green space, apartment complex, attention restoration theory, restorative environments, well-being

## Abstract

Urban green spaces have a positive impact on citizens’ mental health and have contributed to improving their quality of life during the COVID-19 pandemic. In South Korea, where more than 50% of all households live in apartments, apartment-complex landscaping space plays the role of urban green space. This study aimed to investigate the relationships among a perceived restorative environment, restorative experience, life satisfaction, and psychological well-being by conducting a survey between residents living in apartments with landscape space. More specifically, an online survey was conducted from 8 to 15 June 2021 among residents in apartment complexes (500 households or more) located in the capital region in South Korea. We applied partial least squares structural equation modelling (PLS-SEM) using 220 samples to test the causal relationship presented in the conceptual model of this study. The results revealed that residents’ perceptions of the restorative environment of landscape space, including fascination, being away, and coherence had positive effects on restorative attention. Among the restorative environmental factors, the higher the “being away”, the greater the effect on restorative attention. Second, the effects of fascination and coherence on life satisfaction were mediated by restorative attention. Third, restorative attention and life satisfaction significantly influenced psychological well-being. Additionally, life satisfaction acts as a mediator in the relationship between restorative attention and psychological well-being. In summary, this study has theoretical implications, in that it explores the effects of apartment complex landscaping space as urban green spaces on residents’ mental health.

## 1. Introduction

In the past three years, much of the world’s population has been affected by lockdown strategies to prevent the spread of COVID-19. Movement restrictions such as “self-isolation” and “social distancing” had a positive impact on reducing the spread of infectious diseases. Nevertheless, these factors unintentionally have a significant negative effect on people’s physical and psychological health [1,2,3,4]. For example, people often suffer from adult diseases, including cardiovascular diseases and diabetes, and mental disorders, such as depression and stress, have also increased [5,6]. In particular, in the United States, 40.9% of Americans reported experiencing an anxiety disorder or depression due to the pandemic by the end of June 2020 [7]. In addition, 55.8% of people over the age of 15 in Korea experienced anxiety and depression due to the COVID-19 pandemic [8].

In this context, a few studies have shown that the use of urban green space (UGS) mitigates the level of stress and depression that people perceive during the COVID-19 pandemic, while enhancing their physical and mental well-being [9,10]. Green spaces, including parks, roadside trees, and waterfront spaces in the city, provide citizens with opportunities for physical and psychological well-being and social exchanges [11,12,13,14,15]. In other words, the UGS contributes to maintaining a healthy city as a healing environment. In terms of the size of the UGS, the larger it is, the higher citizens’ happiness [16]. For instance, a survey conducted on the residents of major cities in Belgium indicated that residents in areas with well-developed greenery showed relatively high satisfaction with their neighbors and happiness in their life [17].

However, regarding the frequency of UGS use, it was found that accessibility had a more positive impact on the use of UGSs than the scale of UGSs during the lockdown [9,10]. In other words, how often residents are exposed to nature is more important than the scale of UGS. Several studies have highlighted that private gardens played a critical role in experiencing nature during lockdown [18,19,20]. In addition, one study suggested that various types of UGS, including large-scale urban parks and pocket parks, have been established to improve the resilience of urban systems against infectious diseases such as COVID-19 [20]. In particular, they argued that residents should be able to easily access UGSs within walking distance of their home.

Unlike many other countries, Korea can significantly improve its accessibility to UGSs. In Korea, where apartments account for a relatively high proportion of the total housing type, green spaces in apartment complexes are considered an alternative for solving the shortage of UGSs. In fact, 51.1% of all households lived in apartments in 2019, and apartments accounted for 62.3% of all houses in Korea [21]. Apartment complexes consisting of two or more apartments are largely divided into buildings and outdoor spaces, and the latter are subdivided into rest and play spaces, community spaces, and green spaces by function [22]. Initially, the outdoor space of the apartment complex was a form of simply filling the empty space between buildings with greenery. However, in 2000, with the development of underground parking lots and artificial ground-greening technology, the upper part of the parking lot was designed as a landscape space. Recently, as outdoor spaces in apartment complexes have been treated as landscape spaces to enhance comfort and accommodate residents’ activities, the tendency to recognize these spaces as UGSs has been growing [23]. Considering that UGSs contribute to the improvement of citizens’ health and well-being among regions [24] in Korea, where apartment dwellings account for a high proportion of the total housing type, the importance of landscape spaces in apartment complexes is expected to increase. Thus, if we empirically verify the effect of apartment complex landscaping space (ACLS) on people’s mental health, it could contribute to providing a green-space design for residents’ psychological restoration and well-being.

From this perspective, attention restoration theory (ART) is widely used to explain the psychological recovery effect in the natural environment [25,26]. Although several studies based on ART have revealed that the natural environment significantly influences psychological restoration and relaxation [25,26,27,28,29,30,31,32,33], few studies have addressed these relationships from the perspective of ACLS. ART is commonly cited to explain the restorative effects of natural environments, as well as stress-reduction theory (SRT) [34]. Therefore, more studies on the restorative environment of ACLS as UGS are needed. Furthermore, a few researchers have found that the restoration experience through nature has a positive impact on life satisfaction [35] and psychological well-being [32,36]. Thus, it is necessary to identify how the restoration experience through ACLS affects residents’ life satisfaction and psychological well-being. In this context, this study aimed to examine the relationships among the perceived restorative environment, restorative experience, life satisfaction, and psychological well-being of ACLS by residents based on ART. We further attempted to better understand the role of ACLS as a UGS and suggest implications for enhancing residents’ quality of life.

## 2. Theoretical Background and Hypothesis Development

### 2.1. Attention Restoration Theory

In general, the restoration effect of the natural environment has been examined using SRT [37,38] and ART [25,26]. SRT assumes that negative feelings and stress can be reduced when individuals are exposed to an ideal unthreatening nature [37]. In other words, the greenery offered by nature is associated with psychological stability and a calming effect. Similarly, another study found that the increased green level of the roadside environment, such as grass and shrubs, reduced the negative psychological state of the drivers [38].

ART is a theory about the psychological restorative effect of nature and contends that stress can be reduced through exposure to the natural environment [25,26]. In addition, it is explained by two types of attention: directed and involuntary [39]. Directed attention refers to individuals consciously making a certain level of effort to focus on specific information in their daily life. When directed attention accumulates, psychological and physical fatigue increase and eventually become a major cause of stress [40]. In contrast, involuntary attention refers to not consciously making an effort to concentrate on certain information. To recover an individual’s directed attention after burnout, it is necessary to expose them to a restorative environment that stimulates emotional healing [25,26].

According to ART, the restorative environment entails the following four conditions: fascination, being away, extent, and compatibility [25]. First, “being away” refers to the extent to which an individual desires to move away from daily life. Second, “fascination” means that an individual perceives a certain environment as an interesting and attractive place that catches their eye. Third, “extent” refers to the extent to which a sense of space can be recognized and to which the coherence of the elements that make up the environment has been established. Finally, “compatibility” means that the opportunities and constraints given by the environment are compatible with the individual’s goals and disposition.

The perceived restorative scale (PRS) initially developed by Hartig and colleague measures four factors, being away, fascination, coherence, and compatibility, using 26 measurement items [21]. After their study, the PRS was revised and verified by several researchers [27,41,42,43] and is widely used in the restoration assessment of landscape environments. In particular, PRS estimated the restorative value of visual environments to measure five factors (being away, fascination, coherence, scope, and compatibility) [27,42]. In addition, a PRS-11 items scale consisting of 11 items was developed to assess four factors (being away, fascination, coherence, and scope) [41].

A restorative experience means that people experience stress reduction, attention restoration, and energy or vitality recovery through perceived restorativeness. The restoration outcome scale (ROS) uses six items assesses restorative experiences in one’s favorite place [44]. This scale includes three items to measure relaxation and calmness as restorative outcomes [45,46,47].

Meanwhile, previous studies related to ART have demonstrated that the natural environment delivers the benefits of a restoration experience, such as reducing stress or creating positive emotions [27,32,48,49]. People can obtain certain benefits from the natural environment by simply looking at and feeling nature to more actively interact with it by walking and riding a bicycle [50]. For instance, one study found that cities, lakes, mountains, and seas help people regain their focus by creating restorative environments [27]. In addition, exposure to the natural environment can have a positive impact on relieving stress [49], and participation in nature-based recreation shapes restorative experiences that influence emotional well-being [32]. The restorative environment was measured using four items: being away, fascination, coherence, and scope [41].

Hence, based on relevant previous studies, this study proposes the following hypotheses.

**Hypothesis** **1.***The perceived environmental restorativeness of ACLS can have a positive effect on residents’ restorative experience*.

**Hypothesis** **1a.***Being away from the ACLS can have a positive effect on residents’ restorative experience*.

**Hypothesis** **1b.***Perceived fascination with the ACLS can have a positive effect on residents’ restorative experience*.

**Hypothesis** **1c.***Perceived coherence of ACLS can have a positive effect on residents’ restorative experience*.

**Hypothesis** **1d.***The perceived scope of the ACLS can have a positive effect on residents’ restorative experience*.

### 2.2. Life Satisfaction

Life satisfaction has been widely used as an indicator of subjective well-being [51] and is closely associated with a good life [52]. Life satisfaction is measured using multi-item scales, such as the Satisfaction with Life Scale (SWLS), which is a short five-item instrument to measure the cognitive factors of the subjective wellbeing [53,54]. According to the SWLS, life satisfaction can be assessed using the following questions: (1) “How close is my life to the ideal life?”, (2) “How excellent is my life?”, (3) “How satisfied am I with my life?”, (4) “How well do I achieve what I want in my life?”, and (5) “How much would I want to maintain my current life if I were reborn?”

Recently, in the environmental psychology domain, a few studies have examined the relationship between the natural environment and life satisfaction. The frequency of using residential quarter green spaces (RQGS) could positively impact residents’ life satisfaction [38]. Similarly, another study showed that, after greening a schoolyard located in a city, students’ stress levels decreased and psychological well-being improved [48]. Previous studies have demonstrated that restorative environments positively influence life satisfaction [55,56]. Payne et al. [35] verified that university students’ restorativeness had a positive effect on life satisfaction.

Based on previous studies, we assumed that the perception of restorative environments and restorative experiences in ACLS may have a positive effect on residents’ life satisfaction. Therefore, we propose the following hypotheses.

**Hypothesis** **2.***The Perceived environmental restorativeness of the ACLS can have a positive effect on residents’ life satisfaction*.

**Hypothesis** **2a.***Being away from the ACLS can have a positive effect on residents’ life satisfaction*.

**Hypothesis** **2b.***Perceived fascination with the ACLS can have a positive effect on residents’ life satisfaction*.

**Hypothesis** **2c.***Perceived coherence of ACLS can have a positive effect on residents’ life satisfaction*.

**Hypothesis** **2d.***The perceived scope of the ACLS can have a positive effect on residents’ life satisfaction*.

**Hypothesis** **3.***Restorative experiences with ACLS can have a positive effect on residents’ life satisfaction*.

### 2.3. Psychological Well-Being

Happiness is a form of positive emotion that people experience in life and is closely related to various social and economic variables [16]. In terms of UGS, the perception of UGS was considered one of the important variables predicting happiness in life and satisfaction with residential facilities [57]. It has been studied from both the scientific and psychological perspectives since the mid-1980s. Research on happiness has been largely divided into hedonic and eudemonic approaches. The former focuses on the balance between positive and negative emotions regarding happiness, whereas the latter mainly focuses on achieving happiness [58]. Hedonic and eudemonic views have developed into two concepts, subjective and psychological well-being. Subjective well-being refers to the state of well-being that reflects emotions and satisfaction derived from personal experiences [59].

However, psychological well-being is more complicated to define and needs to be explained by multidimensional factors [60]. Psychological well-being was defined using six attributes: self-acceptance, positive relations with others, autonomy, environmental mastery, purpose in life, and personal growth [60]. The higher an individual’s psychological well-being, the higher their self-esteem, positive attitude, and emotions [61].

Restorative experience appears to be a good predictor of psychological well-being during the psychological restoration process in natural environments. A few studies have verified the relationship between nature-based recreational experiences and psychological well-being [32,36]. Another study showed that three restorative environmental factors (being away, fascination, and compatibility) could significantly affect psychological well-being [62]. Life satisfaction is another variable that is regarded as a predictor of psychological well-being. Previous studies have found a positive relationship between life satisfaction and psychological well-being [63,64].

Hence, based on relevant previous studies, this study proposes the following hypotheses.

**Hypothesis** **4.***The restorative experience of ACLS had a positive effect on the psychological well-being*.

**Hypothesis** **5.***Residents’ life satisfaction had a positive effect on their psychological well-being*.

Furthermore, we present the conceptual model of this study in Figure 1.

## 3. Methodology

### 3.1. Data Collection

A questionnaire was developed to test this research model. An online survey with a self-administered questionnaire was conducted twice using a preliminary survey and a final survey from 8 to 15 June 2021 by one of the leading research firms in Korea. The survey targeted residents over the age of 19 who lived in apartment complexes with more than 500 households from the online panels of the research firm, using a purposive sampling method. The complexes were completed within five years in the capital region of South Korea (Seoul, Gyeonggi, and Incheon). All respondents who completed the online survey were paid KRW100 (approximately USD0.90) per min. After verifying the reliability and validity of the measurement items in the questionnaire through a preliminary survey targeting 50 people, we used 220 samples collected from the final survey for this study. In particular, we verified the reliability and validity of the measurement through exploratory factor analysis (EFA). The result of the factor loadings was greater than 0.5, and the Cronbach’s alpha coefficient values were greater than 0.7 (Figure 2).

The minimum sample size required to ensure the test results of this study was determined using the G*POWER 3.0. As a result, the actual power is the statistical power of the test and was 0.95 at a significance level of 0.05, and the minimum sample size was 146, which meets the standard (higher than 0.80, actual power at the significance level 0.05) [65]. Thus, considering that this study had 220 samples, we concluded that the sample collected was suitable for the partial least squares (PLS) methods.

The demographic profiles of the respondents are presented in Table 1. In terms of occupation, office workers accounted for 43.2%, followed by professionals and civil servants (both at 13.2%). Regarding monthly income, 35.9% had a monthly income of more than KRW6,000,000 (approximately USD5004), and 17.7% had a monthly income of KRW4,000,000–5,000,000 (approximately USD3336–4170; KRW1000 = $0.83 as of 7 January 2022).

### 3.2. Measurements

The survey questions were based on a review of the literature and the respondents’ demographic characteristics. We adopted 30 items to measure the seven dimensions. Following previous studies [41,44,53,66], we measured perceived environmental restorativeness, restoration experience, life satisfaction, and psychological well-being (Table 2). Perceived environmental restorativeness was measured using the PRS [41]. This short version of the PRS includes 11 items that comprise 4 subscales of restorative environments based on ART components: being away (BA), fascination (FA), coherence (CO), and scope (SC). Restoration experience was measured using six ROS items [44]. To measure life satisfaction, we used five items from the SWLS [53]. Regarding psychological well-being, eight items were developed based on Diener et al. [66]. All items were measured on a 5-point Likert scale (from ‘1 = strongly disagree’ to ‘5 = strongly agree’ for all measurement items) (Table 2).

### 3.3. Data Analysis Tool

We used the PLS structural equation modeling (PLS-SEM) using SmartPLS version 3.3.9 (SmartPLS GmbH, Oststeinbek, Germany). PLS-SEM has been widely used to estimate the cause–effect relationships among latent variables for the following reasons: (1) compared to other SEM methods, PLS-SEM aims mainly to explore the optimal relations of a number of predictive variables; (2) PLS-SEM determines common factors that best reflect data correlations and chooses the best model in its program; and (3) it uses ordinary least squares (OLS), which can calculate the parameters by maximizing the explainable variance [67,68]. Considering these matters, we assume that PLS-SEM is appropriate for identifying the structural relationships among perceived environmental restorativeness, restorative experience, life satisfaction, and psychological well-being.

## 4. Results

### 4.1. Results of the Measurement Model

The convergent and discriminant validities were verified to identify the construct validity of the constructs presented in the conceptual model. As Table 3 shows, the factor loadings and t values of the items were higher than 0.7 and 1.96 (*p* < 0.05), respectively, indicating that the convergent validity was ensured [69]. Table 4 also shows that the discriminant validity, which refers to the differences between each construct, was confirmed, indicating that the square root of the average variance extracted (AVE) of the construct was higher than the correlation of other constructs [70]. Although the square root of the AVE of the scope was slightly lower than the correlation of coherence and restorative experience, it seemed acceptable; thus, it was concluded that all constructs had discriminant validity. Finally, as Cronbach’s alpha and composite reliability (CR) were above 0.7, the internal consistency validity was confirmed [68,71].

Before examining the research hypotheses, we assessed the overall fit of the conceptual model proposed by R^2^ (variance explained) and communality (ratio of latent variables explained by measurement variables) (Table 5). In general, it has been assumed that the explanatory power is moderate if R^2^ is 0.13–0.26 and relatively high if it is above 0.26 [67]. Table 5 shows that the R^2^ values for restorative experience, psychological well-being, and life satisfaction are higher than 0.23. The model fit, which refers to the square root of the value multiplied by the average of each R^2^ and communality, is considered satisfactory if it is greater than 0.36 [72]. In addition, the PLS model fit can be explained by the standardized root mean square residual (SRMR) and normal fit index (NFI). The former allows the evaluation of the average magnitude of the discrepancy between the observed and expected correlations [73]. In general, if the SRMR value is less than 0.10, the model fit is acceptable [73]. The latter represents the difference between one and the chi-square value of the proposed model divided by the chi-square value of the null model; the closer the NFI value is to 1, the better the model fit [74]. The SRMR and NFI values of this study meet the criterion values of 0.05 and 0.84, respectively; therefore, the model fit of the study is considered acceptable.

### 4.2. Results of the Structural Equation Model

To determine the structural relationships between the variables in the conceptual model, we performed PLS-SEM. The hypothesis test was assessed based on the estimates of the path coefficient and t-value using the bootstrap sampling method. First, as shown in Table 6 and Figure 3, the perceived “fascination” and “coherence” of the restorative environment had a positive impact on the restorative experience (fascination: β = 0.199, t = 2.461 *; coherence: β = 0.158, t = 2.077 *) but did not significantly influence life satisfaction. Second, perceived “being away” had a positive impact on restorative experience (β = 0.580, t = 9.097 ***) but negatively affected life satisfaction (β = −0.245, t = 2.398 *). In sum, perceived “being away” was the most critical factor influencing the relationship between the perception of the restorative environment and the restorative experience. Third, the perceived “scope” of the restorative environment did not affect either the restorative experience or life satisfaction.

In terms of the relationships among the perception of the restorative environment, restorative experience, and life satisfaction, the effects of perceived fascination, being away, and coherence of the restorative environment on life satisfaction were significantly mediated by the restorative experience. Restorative experience completely mediated the relationship between the perceived fascination and coherence of the restorative environment and life satisfaction.

Finally, restorative experience and life satisfaction had positive effects on psychological well-being (restorative experience: β = 0.134, t = 2.115 *; life satisfaction: β = 0.687, t = 13.506 ***). Additionally, life satisfaction played a role as a partial mediator in the effect of the restorative experience on psychological well-being. In other words, the higher the residents’ life satisfaction, the higher their psychological well-being with experience in the ACLS.

## 5. Discussion

We verified the relationships among perceived restorative environments, restorative experience, life satisfaction, and psychological well-being in apartment complex landscaping spaces. Our results indicate that “being away” from everyday life had the greatest influence on restorative experiences. The positive relationship between the perceived restorative environment and the restorative experience theoretically supports previous studies based on ART. In particular, the “fascination” and “coherence” of the restorative environment components are also considered important factors that positively influence the residents’ restorative experience. In other words, for the ACLS to be perceived as a restorative environment, an attractive and balanced design should be considered, in addition to providing a feeling of being out of the ordinary.

However, “being away” had a negative effect on “life satisfaction.” This result is different from a study that investigated whether the perception of restorative environments, including “being away,” has a positive effect on “life satisfaction” [55,56]. Evidence for the experience that ACLS is an adjacent green space seems to have influenced this result. A previous study supported the result that “being away” from a monastery played a role in increasing satisfaction for first-time visitors but did not play an important role for repeat visitors [75]. In this context, it is assumed that “being away” will not have a positive effect on “life satisfaction” for residents who frequently visit the ACLS.

Meanwhile, the perceived “scope” of the restorative environment did not influence restorative attention. This result indicates that the ACLS scale does not affect residents’ restorative experience. In addition, this result also supports previous studies, showing that the scale of UGS does not affect the frequency of UGS if residents are able to use UGSs within walking distance of the residence [9,10]. Mental and physical health conditions were better in residents who had access to green spaces near their homes than in those who did not, indicating that UGS accessibility is important. From this point of view, the ACLS plays the role of a private garden for the residents. Although the ACLS is smaller than urban parks, residents can take a short walk within an apartment complex and experience nature while staying at home. Although ART has been widely explored in the field of urban green space, one of the outstanding findings of this study is that restorative experience plays a significant role in the relationship between restorative environments and life satisfaction.

Restorative experience had a positive impact on life satisfaction, indicating that residents must experience a restorative environment. This result supports previous studies, showing that being in nature can have a positive effect on mental health under stressful circumstances such as COVID-19 [76,77]. However, most restorative environmental components did not directly affect life satisfaction and had an impact on it through restorative experience. In particular, participation time in nature-based activities, rather than visiting periods to restorative environments, affects well-being through restorative experience [32].

Finally, life satisfaction, through restorative experience in ACLS, increased residents’ psychological well-being. Our findings confirm the results of several recent studies, showing that mental health through nature-based activities has a positive effect on psychological well-being [63,78,79,80]. In other words, improving residents’ restorative experiences with ACLS can enhance their quality of life. Therefore, we assumed that ACLS, as a UGS, can play an essential role in enhancing the psychological well-being of residents under stressful circumstances.

## 6. Conclusions

This study verified the psychological effects of the ACLS on residents after the COVID-19 pandemic in Korea using PLS-SEM. Consequently, we found that ACLS had a positive effect on improving residents’ life satisfaction and psychological well-being through restorative experiences. These results may have implications in that the ACLS can be used as a space for enhancing the psychological well-being of residents in situations where social distancing and movement restrictions are imposed because of infectious diseases. Therefore, practitioners need a strategy to improve environmental quality when designing an ACLS based on residents’ restorative experience.

With regard to the restorative environment of landscape space, we suggest that practitioners design apartment landscape spaces with intriguing elements that relieve the monotony of residents’ everyday lives in restorative ways. Considering that landscape spaces in apartment complexes are limited in utilization, it might be difficult to establish various facilities, but we assume that it would be possible to build walking tracks and small-scale exercise facilities. It is important for residents to continuously communicate with nature through leisure activities in the landscaped spaces of apartment complexes in ways that are restorative and improve their psychological well-being.

Although this study has generated insights into the effects of ACLS on residents’ psychological states, several limitations must be considered. Considering that this study conducted a survey during the peak of COVID-19, it is necessary to be cautious when generalizing the results of this study. In future research, it will be meaningful to conduct a survey when residents have fully recovered their daily lives after the end of COVID-19. Furthermore, while this study mainly focused on identifying the causal relationships between restorative environments, restorative experience, life satisfaction, and psychological well-being, respondents’ demographic characteristics such as income level and education were excluded from the analysis. For example, if future studies examine the significant difference between recovery experience and psychological well-being according to income level, richer implications that differ from those of previous studies can be drawn.

Additionally, considering that residents’ life satisfaction positively affects community well-being, it is necessary to consider the effect of apartment landscaping on community resilience. To enhance community resilience, it is necessary to develop leisure programs, especially eco-friendly activities such as gardening, that enable apartment residents to interact and enjoy landscape spaces.

## Figures and Tables

**Figure 1 ijerph-19-10231-f001:**
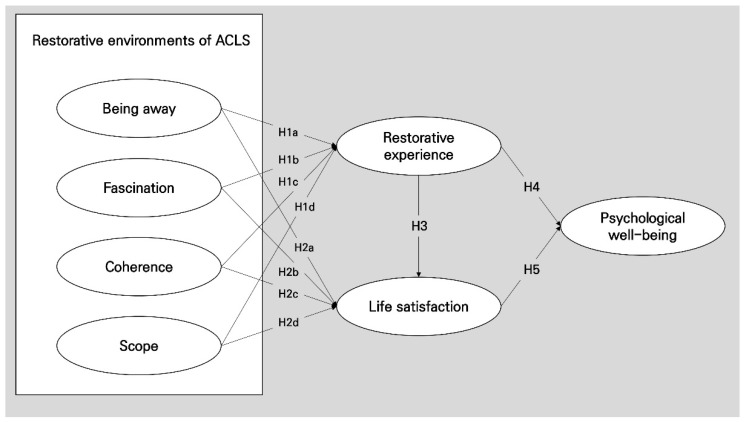
The conceptual model.

**Figure 2 ijerph-19-10231-f002:**
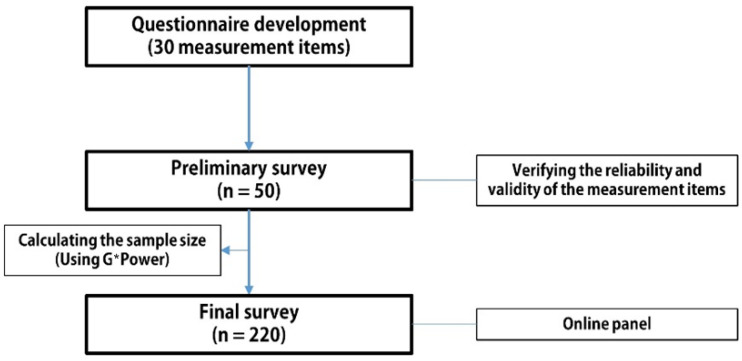
Data-collection procedure.

**Figure 3 ijerph-19-10231-f003:**
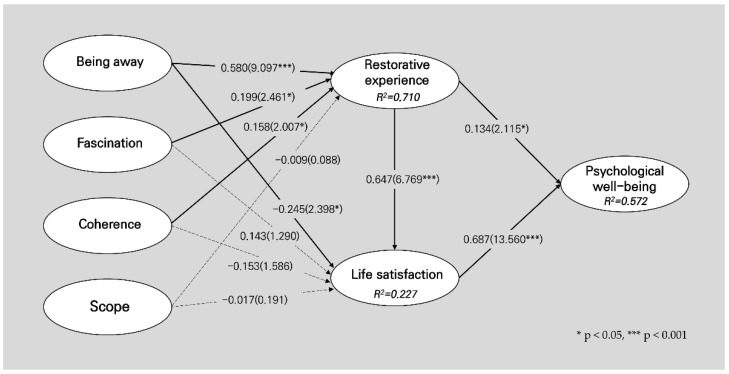
Hypothesis test results.

**Table 1 ijerph-19-10231-t001:** Profile of respondents.

Demographic Traits	Frequency	Percentage (%)
*Sex*		
Male	113	60.5
Female	87	39.5
*Age*		
20–29 years	17	7.7
30–39 years	49	22.3
40–49 years	63	28.6
50–59 years	48	21.8
<60 years	43	19.5
*Occupation*		
Office worker	95	43.2
Professional	29	13.2
Civil servant	29	13.2
Homemaker	20	9.1
Service worker	11	5.0
Student	9	4.1
Laborer	5	2.3
Technician	2	0.9
Other	20	9.1
*Monthly income (KRW1000)*		
Below 1000	3	1.4
1000–2000	4	1.8
2000–3000	24	10.9
3000–4000	33	15.0
4000–5000	39	17.7
5000–6000	38	17.3
6001 or more	79	35.9

**Table 2 ijerph-19-10231-t002:** Measurement items.

Latent Variables		Measurement Items	Reference
Perceived environmental restorativeness (PER)	Being away (BA)	BE1. Places like that are a refuge from nuisances.	Pasini et al. [41]
		BE2. To get away from things that usually demand my attention, I like to go to places like this.	
		BE3. To stop thinking about the things that I must get done, I like to go to places like this.	
	Fascination (FA)	FA1. Places like that are fascinating.	
		FA2. In places like this, my attention is drawn to many interesting things.	
		FA3. In places like this, it is hard to be bored.	
	Coherence (CO)	CO1. There is a clear order in the physical arrangement of places like this.	
		CO2. In places like this, it is easy to see how things are organized.	
		CO3. In places like this, everything seems to have its proper place.	
	Scope (SC)	SC1. That place is large enough to allow exploration in many directions.	
		SC2. In places like this, there are few boundaries to limit my possibility for moving about.	
Restorative experience (RE)		RE1. I feel calmer after being here.	Korpela et al. [44]
		RE2. After visiting this place, I always feel restored and relaxed.	
		RE3. I get new enthusiasm and energy for my everyday routines from here.	
		RE4. My concentration and alertness clearly increase here.	
		RE5. I can forget everyday worries here.	
		RE6. Visiting here is a way of clearing and clarifying my thoughts.	
Life satisfaction (LS)		LS1. In most ways, my life is close to my ideal.	Diener et al. [53]
		LS2. The conditions of my life are excellent.	
		LS3. I am satisfied with my life.	
		LS4. So far, I have gotten the important things I want in life.	
		LS5. If I could live my life over, I would change almost nothing.	
Psychological well-being (PW)		PW1. I lead a purposeful and meaningful life.	Diener et al. [66]
		PW2. My social relationships are supportive and rewarding.	
		PW3. I am engaged and interested in my daily activities.	
		PW4. I actively contribute to the happiness and well-being of others.	
		PW5. I am competent and capable in the activities that are important to me.	
		PW6. I am a good person and live a good life.	
		PW7. I am optimistic about my future.	
		PW8. People respect me.	

**Table 3 ijerph-19-10231-t003:** The result of convergent validity of constructs.

Latent Variables	Items	Loadings	t Value	AVE	CR	Cronbach’s α
Being away (BA)	BA1	0.910	67.271 ***	0.823	0.933	0.892
	BA2	0.921	83.622 ***			
	BA3	0.890	57.294 ***			
Fascination (FA)	FA1	0.900	70.825 ***	0.793	0.920	0.870
	FA2	0.900	55.563 ***			
	FA3	0.871	45.584 ***			
Coherence (CO)	CO1	0.899	54.726 ***	0.796	0.921	0.872
	CO2	0.906	69.886 ***			
	CO3	0.872	43.516 ***			
Scope (SC)	SC1	0.921	57.714 ***	0.790	0.882	0.738
	SC2	0.855	23.112 ***			
Restorative experience (RE)	RE1	0.872	53.824 ***	0.741	0.945	0.930
	RE2	0.851	36.522 ***			
	RE3	0.869	54.500 ***			
	RE4	0.874	52.415 ***			
	RE5	0.877	51.681 ***			
	RE6	0.822	27.448 ***			
Life satisfaction (LS)	LS1	0.821	31.875 ***	0.772	0.944	0.926
	LS2	0.840	40.322 ***			
	LS3	0.859	37.408 ***			
	LS4	0.850	41.222 ***			
	LS5	0.880	54.091 ***			
Psychological well-being (PW)	PW1	0.812	20.057 ***	0.754	0.961	0.953
	PW2	0.835	27.511 ***			
	PW3	0.863	31.637 ***			
	PW4	0.843	23.210 ***			
	PW5	0.906	48.266 ***			
	PW6	0.862	27.819 ***			
	PW7	0.901	46.666 ***			
	PW8	0.917	62.871 ***			

CR: Composite reliability, AVE: Average variance extracted; Significant level: *** *p* < 0.001.

**Table 4 ijerph-19-10231-t004:** The discriminant validity of constructs.

Variables	BA	FA	CO	SC	RE	LS	PW
BA	0.907						
FA	0.705	0.891					
CO	0.610	0.724	0.610				
SC	0.563	0.683	0.563	0.889			
RE	0.811	0.716	0.648	0.571	0.861		
LS	0.348	0.324	0.253	0.242	0.438	0.889	
PW	0.278	0.311	0.208	0.198	0.442	0.746	0.879

Footnote: The square root of the AVE is marked as italic type. FA: Fascination, BA: Being away, CO: Coherence, SC: Scope, RE: Restorative experience, PW: Psychological well-being, LS: Life satisfaction.

**Table 5 ijerph-19-10231-t005:** Overall model fit.

	Restorative Experience	Life Satisfaction	Psychological Well-Being
R^2^	0.710	0.227	0.572
Communality	0.742	0.772	0.723
Model Fit	0.613		

**Table 6 ijerph-19-10231-t006:** Hypothesis analytics.

Hypothesis Path	Coefficient	S.D.	t Value
Being away → Restorative experience	0.580	0.064	9.097 ***
Being away → Life satisfaction	−0.245	0.102	2.398 *
Fascination → Restorative experience	0.199	0.081	2.461 *
Fascination → Life satisfaction	0.143	0.111	1.290
Coherence → Restorative experience	0.158	0.076	2.077 *
Coherence → Life satisfaction	−0.153	0.096	1.586
Scope → Restorative experience	−0.009	0.101	0.088
Scope → Life satisfaction	−0.017	0.091	0.849
Restorative experience → Life satisfaction	0.647	0.096	6.769 ***
Restorative experience → Psychological well-being	0.134	0.064	2.115 *
Life satisfaction → Psychological well-being	0.687	0.051	13.560 ***

* *p* < 0.05, *** *p* < 0.001.

## Data Availability

Not applicable.

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
