# Peer review of "How Can Apartment-Complex Landscaping Space Improve Residents’ Psychological Well-Being?: The Case of the Capital Region in South Korea"

_ijerph, 2022, doi:10.3390/ijerph191610231_

Round 1

Reviewer 1 Report

(1) The big problem of this manuscript is authors must clarify all the concepts used. Although the concepts have been mentioned in section 2, they are far from clear and logical. Concepts in the manuscript are quite blurring.

(2) Also, in Methodology section, the references for the scales were quoted in the tables, but not explained in the text. Who developed the scales, how many items of the scales in their original form, and the reliability, the construct…? They should be stated and appropriated quoted in the text.

(3) About the sample size, how authors justify them enough to test the hypotheses?

(4) The introduction section, the research problem was not properly explained and proposed.

(5) Lines 112-117, we don’t cite papers use “[26]” as the subject, names of the authors of the literature are recommended. This issue happens throughout the manuscript.

(6) Lines 134-146, “happiness” is not tested as a variable, but explained a lot in the text. Authors should focus on the concepts relevant with their content.

(7) Lines 196-205, are samples of “preliminary survey” used in the study? Be part of 220?

(8) The results are also not well presented.

Author Response

Point 1: The big problem of this manuscript is authors must clarify all the concepts used. Although the concepts have been mentioned in section 2, they are far from clear and logical. Concepts in the manuscript are quite blurring.

Response 1: Thank you for your thoughtful comment. We have revised the theoretical background in accordance with the logic you proposed. Please check the section 2 (Theoretical background and hypothesis development).

Point 2: Also, in Methodology section, the references for the scales were quoted in the tables, but not explained in the text. Who developed the scales, how many items of the scales in their original form, and the reliability, the construct…? They should be stated and appropriated quoted in the text.

Response 2: Thank you for your thoughtful comment. We have added more explanations about measurement items of the scales in the section 3.2 (Measurements).

(Line 249-260) “The survey questions were based on a review of the literature and respondents’ demographic characteristics. We adopted 30 items to measure the seven dimensions. Fol-lowing previous studies [39,42,51,64] we measured perceived environmental restorative-ness, restoration experience, life satisfaction, and psychological wellbeing (Table 2). Perceived environmental restorativeness was measured using PRS [39]. This short version of the PRS includes 11 items that comprise four subscales of restorative environments based on ART components: being away (BA), fascination (FA), coherence (CO), and scope (SC). Restoration experience was measured using six ROS items [42]. To measure life satisfaction, we used five items from the SWLS [51]. Regarding psychological well-being, eight items were developed based on Diener et al. [64]. All items were measured on a 5-point Likert scale (from ‘1=strongly disagree’ to ‘5=strongly agree’ for all measurement items) (Table 2).”

Point 3: About the sample size, how authors justify them enough to test the hypotheses?

Response 3: Following your advice, we have added the sentence about the sample size.

(Line 234-239) “We determined the minimum sample size required to ensure the test results for this study using G*POWER 3.0. As a result, the actual power, which means the statistical power of the test, was 0.95 at a significance level of 0.05, and the minimum sample size was 146, which meets the standard (higher than 0.80, actual power at the significance level 0.05) [63]. Thus, considering that this study had 220 samples, we concluded that the sample we collected was suitable for the partial least squares (PLS) methods.”

Point 4: The introduction section, the research problem was not properly explained and proposed.

Response 4: Thank you for your thoughtful comment. We have revised and clearly presented research questions in the section 1.

(Line 81-94) “Although several previous studies based on attention restoration theory (ART) have revealed that the natural environment significantly influences psychological restoration and relaxation [25-29], few studies have addressed these relationships from the perspective of apartment complex landscaping spaces. ART is commonly cited to explain restorative effects of natural environments as well as stress reduction theory (SRT) [30]. Therefore, more study on the restorative environments of apartment complex landscaping space (ACLS) as an UGS is needed. Furthermore, a few researchers found that the restoration experience through nature has a positive impact on life satisfaction [31] and psychological well-being [32,33]. Thus, it is necessary to identify how restoration experience through ACLS affects residents’ life satisfaction and psychological well-being. In this context, this study aims to examine the relationships between the perceived restorative environment, restorative experience, life satisfaction, and psychological well-being of ACLS by residents based on ART. We further attempt to better understand the role of ACLS as an UGS and suggest implications for enhancing residents’ quality of life.”

Point 5: Line 112-117, we don’t cite papers use “[26]” as the subject, names of the authors of the literature are recommended. This issue happens throughout the manuscript.

Response 5: Following your advice, we have changed citation style and modified the sentence.

(Line 59) “Ugolini et al. [20] suggested...”/(Line 100) “Jiang et al. [35] found that…”/ (Line 125) “Betro [25] and Purcell et al. [40] presented…”/(Line 127) “Pasini et al. [39] developed…”/(Line 131-132) “Korpela et al. [42] presented…”/(Line 139) “Betro [25] revealed…”/(Line 143) “Pasini et al. [39] measured…”/(Line 161) “Diener et al. [51] to measure…”/(Line 168) “Huang [35] suggested…”/(Line 169) “Kelz et al. [46] showed…”/(Line 172) “Payne et al. [31] verified…”/(Line 202) “Ryff [58] defined…”/(Line 209) “Yusli et al. [60] demonstrated…”/(Line 362) “Korpela et al. [33] suggested…”

Point 6: Line 134-146, “happiness” is not tested as a variable, but explained a lot in the text. Authors should focus on the concepts relevant with their content.

Response 6: Thank you for your in-depth review of theoretical background section. We have added the theoretical background of happiness.

(Line 190-193) "Happiness is a form of positive emotion that people experience in life and is closely related to different social and economic variables [16]. In terms of UGS, perception of UGS was considered one of the important variables predicting happiness in life and satisfac-tion with residential facilities [55]."

Point 7: Line 196-205, are samples of “preliminary survey” used in the study? Be part of 220?

Response 7: We conducted preliminary and final survey separately. So, we did not use sample of preliminary survey for the final survey. Regarding this, we have added the explanations in the section 3.1 (Data collection).

(Line 232-233) “After verifying the reliability and validity of the measurement items on the questionnaire through a preliminary survey targeting 50 people, we used a total of 220 samples collected from the final survey for this study (Figure 2).”

Point 8: The results are also not well presented.

Response 8: Thank you for your thoughtful comment. We have revised the results. Please check the section 4.

Reviewer 2 Report

Dear Authors, I have reviewed the manuscript "How Can Apartment Landscape Space Improve Residents' Psychological Well-being as a Resilience Strategy to Covid-19?: The Case of Korea", which aimed to "investigate the relationships of perceived restorative environment, restorative experience, life satisfaction, and psychological well-being by conducting a survey on residents living in apartments with landscape space". Prior to further processing, the following observations and comments should be addressed:

1. Include a paragraph in the introduction about happiness.

2. Explain why you developed a preliminary and final survey in the methodology. Additionally, when you start writing the methodology, it is necessary to clearly describe the design and focus of your research.

3. It is necessary to increase the citations in statements made about the perceived and restored scope in the discussion.

4. In the conclusions it is necessary to increase a paragraph on the limitations of your manuscript and future studies based on your results.

Author Response

Dear Authors, I have reviewed the manuscript "How Can Apartment Landscape Space Improve Residents' Psychological Well-being as a Resilience Strategy to Covid-19?: The Case of Korea", which aimed to "investigate the relationships of perceived restorative environment, restorative experience, life satisfaction, and psychological well-being by conducting a survey on residents living in apartments with landscape space". Prior to further processing, the following observations and comments should be addressed:

Point 1: Include a paragraph in the introduction about happiness.

Response 1: Following your advice, we have added explanations about happiness in the section 2.3.

(Line 190-193) "Happiness is a form of positive emotion that people experience in life and is closely related to different social and economic variables [16]. In terms of UGS, perception of UGS was considered one of the important variables predicting happiness in life and satisfaction with residential facilities [55]."

Point 2: Explain why you developed a preliminary and final survey in the methodology. Additionally, when you start writing the methodology, it is necessary to clearly describe the design and focus of your research.

Response 2: Thank you for your thoughtful comment. We should verify the reliability and validity of the measurement items on the questionnaire through a preliminary survey before conducting the final survey.

(Line 231-232) "After verifying the reliability and validity of the measurement items on the questionnaire through a preliminary survey targeting 50 people, we used a total of 220 samples collected from the final survey for this study.”

Point 3: It is necessary to increase the citations in statements made about the perceived and restored scope in the discussion.

Response 3: Thank you for your thoughtful comment. We have added the citations about relationship restorative environment and life satisfaction in the section 5.

(Line 336-343) “However, “being away” had a negative effect on “life satisfaction.” This result is different from a study that investigated whether the perception of restorative environments, including “being away,” has a positive effect on “life satisfaction” [53,54]. The evidence for the experience that ACLS is an adjacent green space seems to have influenced this result. For example, Ouellette et al. [71] argued that “being away” from monastery played a role in increasing satisfaction for first-time visitors but did not play an important role for repeat visitors. In this context, it is assumed that “being away” will not have a positive effect on “life satisfaction” for residents who frequently visit the ACLS”.

Point 4: In the conclusions it is necessary to increase a paragraph on the limitations of your manuscript and future studies based on your results.

Response 4: With regard to your valuable suggestion, we have improved the section 6 in order to address your concerns.

(Line 389-400) “Although this work has generated insights into the effects of ACLS on residents’ psycho-logical states, several limitations must be considered. Considering that this study conducted a survey during the peak of COVID-19, it is necessary to be cautious in generalizing the results of this study. In future research, it will be meaningful to conduct a survey at a time when residents have fully recovered their daily lives after the end of COVID-19. Furthermore, while this study mainly focused on identifying the causal relationships be-tween restorative environments, restorative experience, life satisfaction, and psychological well-being, respondents’ demographic characteristics such as income level and education were excluded from the analysis. For example, if future studies examine the significant difference between recovery experience and psychological well-being according to income level, richer implications that differ from those of previous studies could be drawn.”

Reviewer 3 Report

This study evaluates the impact of urban green space on citizens’ mental health. This study investigated the relationships of perceived restorative environment, restorative experience, life satisfaction, and psychological well-being by conducting a survey on residents living in apartments with landscape space. The authors utilized Smart PLS (partial least squares) using 220 samples to test the causal relationship presented on the conceptual model of this study. The results revealed the effects of apartment landscape space as an urban green space on residents’ mental health. There are several concerns about how the research was organized and how the questionnaire was carried out. The paper's title appears to be inappropriate. Overall, then, the paper has some significant merits. I do however also see some significant problems. I comment below on some more general issues that arose for me as I read the paper and which the authors would need to address in major revisions, if they are invited to revise and resubmit the paper.

One general issue has to do with the authors' representation of their experiment versus what they have actually done. In the title and elsewhere, they refer to "Covid - 19" and “Korea”; however, I found this study is not necessarily related to Covid -19. In the questionnaire the authors used, no question was about Covid – 19 ,special life style or unnormal life caused by Covid – 19.

The authors use Korea in the title, but their only surveyed the Seoul metropolitan area. So, I think it is not suitable to say Korea in the title.

Detailed comments:

        Perhaps should not include “Covid -19”, or add content to make the title suitable.

        Also, use “Seoul metropolitan area, Korea” instead of “Korea”.

        Line 19. PLS (partial least squares) to partial least squares (PLS).

        Line 112 and so on. “[26] proposed the perceived restorative” this citation style seems strange, please check.

l  Line 134-146. This part is not clear. Please define the “Happiness”, “subjective well-being”, “psychological well-being”, “subjective well-being”, “psycho-logical well-being”, “life satisfaction” and clarify their differences in this study.

l  Line148-157. The authors mentioned attention restoration theory, but why did not mention stress reduction theory?

l  Line 167-191. This whole part is too long and redundant. Please make this part concise. Maybe a table is better.

l  Figure 1. The relationship between “restorative experience” and “life satisfaction”, also their relationship with “psychology well-being” should be explained very clearly.

l  Please provide ethics statement with number.

l  Line 202. Please explain how you avoid volunteer effect.

l  Please describe the whole survey very clear. Maybe a table or a flow chart is better.

l  Table 1.non of these questions related to Covid-19.

l  Line 214-224. Please show the software that authors used to conduct data analysis.

l  Table 2. It is better to move this table to methodology.

l  Table 2. “gender” to “sex”.

l  Line 239. Pease use full spell of AVE when it appeared first time.

l  Line 264. “β=--0.245” to “β=-0.245”.

l  Line 262-264. The authors mentioned “Second, the perceived ‘being away’ had a positive impact on the restorative experience (β=0.580, t=9.097***) but negatively affected life satisfaction (β=--0.245, t=2.398*).”. Please discuss this result in the discussion part.

l  Line 287-289. “This result might reflect the increase in stress and the desire for going out and participating in outdoor activities due to the government’s ‘social distancing’ policy after the outbreak of COVID-19.”. This may be right. But how did the authors get this from a questionnaire with nothing about the social distancing’ policy?

l  Line 298-301. “It was found that mental and physical health conditions were better in residents who had access to green spaces near their homes than in those who did not, which means that UGS accessibility is important.” Where is this “physical health” come from, I did not found in your results part.

l  Line 325. “enhancing psychological resilience of residents in a situation where social distancing and movement re strictions are imposed due to COVID-19.” Same question as above.

Author Response

This study evaluates the impact of urban green space on citizens’ mental health. This study investigated the relationships of perceived restorative environment, restorative experience, life satisfaction, and psychological well-being by conducting a survey on residents living in apartments with landscape space. The authors utilized Smart PLS (partial least squares) using 220 samples to test the causal relationship presented on the conceptual model of this study. The results revealed the effects of apartment landscape space as an urban green space on residents’ mental health. There are several concerns about how the research was organized and how the questionnaire was carried out. The paper's title appears to be inappropriate. Overall, then, the paper has some significant merits. I do however also see some significant problems. I comment below on some more general issues that arose for me as I read the paper and which the authors would need to address in major revisions, if they are invited to revise and resubmit the paper.

One general issue has to do with the authors' representation of their experiment versus what they have actually done. In the title and elsewhere, they refer to "Covid - 19" and “Korea”; however, I found this study is not necessarily related to Covid -19. In the questionnaire the authors used, no question was about Covid – 19 ,special life style or unnormal life caused by Covid – 19.

The authors use Korea in the title, but their only surveyed the Seoul metropolitan area. So, I think it is not suitable to say Korea in the title.

Point 1: Perhaps should not include “Covid -19”, or add content to make the title suitable. Also, use “Seoul metropolitan area, Korea” instead of “Korea”.

Response 1: As you recommended, we have deleted “Covid-19”. Also, we have revised the title.

(Line 2-4) How Can Apartment Complex Landscaping Space Improve Residents’ Psychological Well-being?: The Case of the Capital Region in South Korea

Point 2: Line 19. PLS (partial least squares) to partial least squares (PLS).

Response 2: Following your advice, we have revised it.

(Line 238-239) “Thus, considering that this study had 220 samples, we concluded that the sample we collected was suitable for the partial least squares (PLS) methods.”

Point 3: Line 112 and so on. “[26] proposed the perceived restorative” this citation style seems strange, please check.

Response 3: Following your advice, we have changed citation style and modified the sentence.

(Line 59) “Ugolini et al. [20] suggested...”/(Line 100) “Jiang et al. [35] found that…”/ (Line 125) “Betro [25] and Purcell et al. [40] presented…”/(Line 127) “Pasini et al. [39] developed…”/(Line 131-132) “Korpela et al. [42] presented…”/(Line 139) “Betro [25] revealed…”/(Line 143) “Pasini et al. [39] measured…”/(Line 161) “Diener et al. [51] to measure…”/(Line 168) “Huang [35] suggested…”/(Line 169) “Kelz et al. [46] showed…”/(Line 172) “Payne et al. [31] verified…”/(Line 202) “Ryff [58] defined…”/(Line 209) “Yusli et al. [60] demonstrated…”/(Line 362) “Korpela et al. [33] suggested…”

Point 4: Line 134-146. This part is not clear. Please define the “Happiness”, “subjective well-being”, “psychological well-being”, “subjective well-being”, “psychological well-being”, “life satisfaction” and clarify their differences in this study.

Response 4: Thank you for your in-depth review of theoretical background section. We have revised the theoretical background in accordance with your advice. Please check the section 2 (Theoretical background and hypothesis development).

Point 5: Line148-157. The authors mentioned attention restoration theory, but why did not mention stress reduction theory?

Response 5: Thank you so much for raising the issue. As you suggest, we have addressed the stress reduction theory.

(Line 97-102) “In general, the natural environment’s restoration effect has been examined by SRT [34,35] and ART [29,36]. SRT assumes that negative feelings and stress can be reduced when individuals are exposed to an ideal, unthreatening nature [34]. In other words, the greenery offered by nature is associated with psychological stability and a calming effect. Jiang et al. [35] found that the higher the green level of the roadside environment, such as grass and shrubs, the lower the negative psychological state of drivers.”

Point 6: Line 167-191. This whole part is too long and redundant. Please make this part concise. Maybe a table is better.

Response 6: As you recommended, we have revised this part more clearly in the section 2. Please check the line 147-156, 177-188, and 216-218.

Point 7: Figure 1. The relationship between “restorative experience” and “life satisfaction”, also their relationship with “psychology well-being” should be explained very clearly.

Response 7: We appreciate for your comment. We have revised the relationships of restorative experience, life satisfaction, and psychology well-being based on the results of previous studies.

(Line 166-173) “Recently, in the environmental psychology domain, a few studies have examined the relationship between the natural environment and residents’ life satisfaction. Jiang and Huang [35] suggested that the frequency of use of residential quarter green spaces (RQGS) could have a positive impact on residents’ life satisfaction. Kelz et al. [46] showed that after greening a schoolyard located in a city, students’ stress levels decreased, while psychological well-being improved. A few previous studies have demonstrated that restorative environments positively influence life satisfaction [53,54]. Payne et al. [31] verified that university students’ restorativeness has a positive effect on life satisfaction.”

(Line 206-213) “Restorative experience appears to be a good predictor of psychological well-being in the psychological restoration process in natural environments. A few studies have verified the relationship between nature-based recreational experiences and psychological well-being [32,33]. In addition, Yusli et al. [60] demonstrated that three restorative environmental factors (being away, fascination, and compatibility) significantly affect psychological well-being. Another variable regarded as one of the predictors of psychological well-being is life satisfaction. Previous studies have found a positive relationship between life satisfaction and psychological well-being [61,62].”

Point 8: Please provide ethics statement with number.

Response 8: Regarding the ethical code (or ethical approval file) for the research, we think it is not applicable to our paper. Since the survey we conducted by a professional research firm having a number of online panels, this study is not required the approval of the Institutional Review Board (IRB). We have reported this matter to the assistant editor.

Point 9: Line 202. Please explain how you avoid volunteer effect.

Response 9: We appreciate for your comment. We collected data (sample) from a paid online survey firm having approximately 18 million online panel after screening several criteria such as the year the apartment was built, the number of households in the apartment, and whether the apartment has landscaping space.

Point 10: Please describe the whole survey very clear. Maybe a table or a flow chart is better.

Response 10: Following your advice, we have added figure 2. Please check the line 240.

Point 11: Table 1.non of these questions related to Covid-19.

Response 11: Following your advice, we have confirmed that there is no question about COVID-19 in our survey questionnaire. Thus, we have removed COVID-19 from “title” and revised the contents for COVID-19 in the “Discussion” and “Conclusion”.

Point 12: Line 214-224. Please show the software that authors used to conduct data analysis.

Response 12: Following your advice, we have revised it.

(Line 263-264) “To verify the hypotheses proposed in this study, we used the PLS structural equation modelling (PLS-SEM) using SmartPLS version 3.3.9.”

Point 13: Table 2. It is better to move this table to methodology.

Response 13: Thank you for your thoughtful comment. Table 2 has been moved from section 4.1. (Sample description) to section 3.1. (Data collection). Please check the line 247.

Point 14: Table 2. “gender” to “sex”.

Response 14: Following your advice, we have changed “gender” to “sex” in Table 1. Please check the line 247.

Point 15: Line 239. Pease use full spell of AVE when it appeared first time.

Response 15: Following your advice, we have revised it. Please check the line 280.

Point 16: Line 264. “β=--0.245” to “β=-0.245”.

Response 16: Following your advice, we have changed “β=--0.245” to “β=-0.245”. Please check the line 306.

Point 17: Line 262-264. The authors mentioned “Second, the perceived ‘being away’ had a positive impact on the restorative experience (β=0.580, t=9.097***) but negatively affected life satisfaction (β=--0.245, t=2.398*).”. Please discuss this result in the discussion part.

Response 17: Thank you for your thoughtful comment. We have added more explanations in the discussion.

(Line 336-343) “However, “being away” had a negative effect on “life satisfaction.” This result is different from a study that investigated whether the perception of restorative environments, including “being away,” has a positive effect on “life satisfaction” [53,54]. The evidence for the experience that ACLS is an adjacent green space seems to have influenced this result. For example, Ouellette et al. [71] argued that “being away” from monastery played a role in increasing satisfaction for first-time visitors but did not play an important role for repeat visitors. In this context, it is assumed that “being away” will not have a positive effect on “life satisfaction” for residents who frequently visit the ACLS.”

Point 18: Line 287-289. “This result might reflect the increase in stress and the desire for going out and participating in outdoor activities due to the government’s ‘social distancing’ policy after the outbreak of COVID-19.”. This may be right. But how did the authors get this from a questionnaire with nothing about the ‘social distancing’ policy?

Response 18: Thank you for your thoughtful comment. We have revised it. Please check the section 5 (Discussion).

Point 19: Line 298-301. “It was found that mental and physical health conditions were better in residents who had access to green spaces near their homes than in those who did not, which means that UGS accessibility is important.” Where is this “physical health” come from, I did not found in your results part.

Response 19: Thank you for your advice. We did not deal with “physical health” in this study, and thus we have removed “physical health” from the sentence.

Point 20: Line 325. “enhancing psychological resilience of residents in a situation where social distancing and movement re strictions are imposed due to COVID-19.” Same question as above.

Response 20: Thank you so much for your suggestion. We have revised it.

(Line 376-378) “These results may have implications in that the ACLS can be used as a space for enhancing the psychological well-being of residents in situations where social distancing and movement restrictions are imposed due to infectious diseases.”

Reviewer 4 Report

The authors need to present the theoretical foundation regarding perception and restorative experience as more concrete manners.

Also, life satisfaction and well being needs to be separated with more in-depth review of literature.

Why Seoul, GY, and INC are selected? The authors need to elaborate data collection procedure.

What about the data cleaning procedure? What about valid rate of sample?

The authors need to present own definition of variables in the measurement part,

How goodness of fit assessed? It needs to be presented at data analysis section.

Also, in the results goodness of fit indices, and results of convergent validity and discriminant validity needs to be presented.

The authors need to strengthen theoretical implication of this research based on ART. It should be strengthened. 

The authors also need to present why this research is necessary? What could become the research gap? What can be contribution of this research?

Author Response

Point 1: The authors need to present the theoretical foundation regarding perception and restorative experience as more concrete manners. Also, life satisfaction and well-being needs to be separated with more in-depth review of literature.

Response 1: Thank you for your advice. We have revised the theoretical background including perceived restorative environment, restorative experience, life satisfaction and psychological well-being. Please check the section 2.

Point 2: Why Seoul, GY, and INC are selected? The authors need to elaborate data collection procedure.

Response 2: Thank you for your thoughtful comment. The metropolitan area, including Seoul, Gyeonggi, and Incheon, has the largest number of large-scale apartment complexes with more than 1,000 households in Korea. In addition, according to Korean legislation, for residential complexes with more than 300 households, 3/10 of the total area of the complex should be green. Considering the above, thus, this study conducted a survey on residents of newly built apartments with more than 500 households in the metropolitan area including Seoul, Gyeonggi, and Incheon.

Point 3: What about the data cleaning procedure? What about valid rate of sample?

Response 3: Thank you for your thoughtful comment. As we described in the 3.1. (Data collection), we used the final data received from a professional research firm, which handled missing data and duplication observation as well as conducted a survey.

Point 4: The authors need to present own definition of variables in the measurement part.

Response 4: Thank you for your thoughtful comment. We have revised it. Please check the Section 3.2.

(Line 249-260) “The survey questions were based on a review of the literature and respondents’ demographic characteristics. We adopted 30 items to measure the seven dimensions. Following previous studies [39,42,51,64] we measured perceived environmental restorativeness, restoration experience, life satisfaction, and psychological well-being (Table 2). Perceived environmental restorativeness was measured using PRS [39]. This short version of the PRS includes 11 items that comprise four subscales of restorative environments based on ART components: being away (BA), fascination (FA), coherence (CO), and scope (SC). Restoration experience was measured using six ROS items [42]. To measure life satisfaction, we used five items from the SWLS [51]. Regarding psychological well-being, eight items were developed based on Diener et al. [64]. All items were measured on a 5-point Likert scale (from ‘1=strongly disagree’ to ‘5=strongly agree’ for all measurement items) (Table 2).”

Point 5: How goodness of fit assessed? It needs to be presented at data analysis section.

Response 5: Thank you for bringing the issue. However, regarding it, we addressed that model fit in the section 4.2.

(Line 293-295) “In addition, the model fit, which refers to the square root of the value multiplied by the average of each R2 and communality, is considered satisfactory if it is higher than 0.36 [70].”

Point 6: Also, in the results goodness of fit indices, and results of convergent validity and discriminant validity needs to be presented.

Response 6: Before presenting the result of goodness of fit, we explained about convergent validity and discriminate validity of the measurement model in line 275-281. We kindly ask you to read section 4.1. again.

Point 7: The authors need to strengthen theoretical implication of this research based on ART. It should be strengthened.

Response 7: Thank you for your thoughtful comment. We have revised it.

(Line 353-356) “Although ART has been widely explored in the field of urban green space, one of the out-standing findings of this study is that restorative experience significantly plays a critical role in the relationship between restorative environments and life satisfaction.”

Point 8: The authors also need to present why this research is necessary? What could become the research gap? What can be contribution of this research?

Response 8: Thank you for your thoughtful comment. One of the outstanding findings of this study revealed that ACLS plays an important role in providing urban greenery to residents. Thus, we expect that this study is meaningful in that it suggests a direction in which ACLS can improve the quality of life of residents. Please check the line 366-371.

Reviewer 5 Report

This is an interesting paper looking at perceived restorative environment with an East Asian case. Generally, the research design is reliable. The following are just for your reference.

The preliminary survey’s intention, findings as well as insights for the final survey should be articulated, which will bring the audience more context.

There are typos of words, and punctuation (for example, Table. 4, “220”and “<” in Table 2) and inappropriate expressions.

The findings seem within expectation, comparing to those similar studies relevant to green spaces and parks. Any new findings, even subtle? Maybe further analysis of the interviewees would imply something new, such as correlation between income/career and wellbeing/restorative capability?  Do you consider self-selection or other confounding factors, such as the wealthier people prefer and can afford greener environment?  If these questions are addressed, the contribution would be more meaningful and important.  

what particular new theoretical, methodological and practical contributions the study makes need to be clearly spelled out in the introduction and conclusion.

Author Response

Point 1: This is an interesting paper looking at perceived restorative environment with an East Asian case. Generally, the research design is reliable. The following are just for your reference.

Response 1: We greatly appreciate your encouraging comments. We have revised our manuscript according to your suggestion below.

Point 2: The preliminary survey’s intention, findings as well as insights for the final survey should be articulated, which will bring the audience more context.

Response 2: We appreciate your keen and thoughtful observation. As you suggested, we have revised the section 3.1.

(Line 231-233) “After verifying the reliability and validity of the measurement items on the questionnaire through a preliminary survey targeting 50 people, we used a total of 220 samples collected from the final survey for this study (Figure 2).”

Point 3: There are typos of words, and punctuation (for example, Table. 4, “220”and “<” in Table 2) and inappropriate expressions.

Response 3: Following your advice, we have revised it. Please check the line 247 and 286.

Point 4: The findings seem within expectation, comparing to those similar studies relevant to green spaces and parks. Any new findings, even subtle? Maybe further analysis of the interviewees would imply something new, such as correlation between income/career and wellbeing/restorative capability?  Do you consider self-selection or other confounding factors, such as the wealthier people prefer and can afford greener environment?  If these questions are addressed, the contribution would be more meaningful and important.  

Response 4: With regard to your valuable suggestion, we have improved the entire conclusion section in order to address your concerns.

(Line 389-400) “Although this work has generated insights into the effects of ACLS on residents’ psycho-logical states, several limitations must be considered. Considering that this study conducted a survey during the peak of COVID-19, it is necessary to be cautious in generalizing the results of this study. In future research, it will be meaningful to conduct a survey at a time when residents have fully recovered their daily lives after the end of COVID-19. Furthermore, while this study mainly focused on identifying the causal relationships between restorative environments, restorative experience, life satisfaction, and psychological well-being, respondents’ demographic characteristics such as income level and education were excluded from the analysis. For example, if future studies examine the significant difference between recovery experience and psychological well-being according to income level, richer implications that differ from those of previous studies could be drawn.”

Point 5: What particular new theoretical, methodological and practical contributions the study makes need to be clearly spelled out in the introduction and conclusion.

Response 5: Thank you for your thoughtful comment. We have added more explanations about it. Please check the line 352-355 and 375-377.

(Line 353-356) “Although ART has been widely explored in the field of urban green space, one of the out-standing findings of this study is that restorative experience significantly plays a critical role in the relationship between restorative environments and life satisfaction.”

(Line 376-378) “These results may have implications in that the ACLS can be used as a space for enhancing the psychological well-being of residents in situations where social distancing and movement restrictions are imposed due to infectious diseases.”

Round 2

Reviewer 1 Report

The authors have improved the manuscript.

Lines 228-230: The method for delivering the online survey should be stated in detail. Did the survey cover all targeted residents? Or there is specific sampling method to ensure the samples are a smaller set of the population?

Although I am not qualified to judge about the language, it is suggested to make further improvement.

Author Response

Point 1: The authors have improved the manuscript.

Lines 228-230: The method for delivering the online survey should be stated in detail. Did the survey cover all targeted residents? Or there is specific sampling method to ensure the samples are a smaller set of the population?

Although I am not qualified to judge about the language, it is suggested to make further improvement.

Response 1: Thank you for your thoughtful comment. We have added more explanations about a sampling method in the section 3.1 (Data collection).

(Line 229-232) The survey targeted residents over the age of 19 who lived in apartment complexes with more than 500 households from online panels of the research firm, using a purposive sampling method.

Reviewer 2 Report

Dear authors, I have revised manuscript "How Can Apartment Landscape Space Improve Residents' Psychological Well-being as a Resilience Strategy to Covid-19? : The Case of Korea" for the second time. Thank you for the work done, all comments have been considered. This is a better version of the manuscript.

Author Response

Point 1: Dear authors, I have revised manuscript "How Can Apartment Landscape Space Improve Residents' Psychological Well-being as a Resilience Strategy to Covid-19? : The Case of Korea" for the second time. Thank you for the work done, all comments have been considered. This is a better version of the manuscript.

Response 1: We greatly appreciate your insightful and helpful comments on our manuscript.

Reviewer 3 Report

The manuscript is well revised. There are anyway some points can be improved.

l  Line 59 and so on. The citation style is still kind of not appropriate. Such as “Ugolini et al. [20]”. I recommend authors move the “[20]” to the end of the sentence.

l   Line 81-87. The low number of studs on this term is not reasonable. Authors should explain why this study should be done, from the perspective of how people can benefit from it.

l  Line 232-234. Please explain how authors “verifying the reliability and validity of the measurement items on the questionnaire through a preliminary survey targeting 50 people”. Describe the detail, what kind of method and so on.

l     

Author Response

The manuscript is well revised. There are anyway some points can be improved.

Point 1: Line 59 and so on. The citation style is still kind of not appropriate. Such as “Ugolini et al. [20]”. I recommend authors move the “[20]” to the end of the sentence.

Response 1: Following your advice, we have changed citation style and modified the sentence.

(Line 59-62) In addition, one study suggested that various types of UGS, including large-scale urban parks and pocket parks, have been established to improve the resilience of urban systems against infectious diseases such as COVID-19 [20].

(Line 105-106) Similarly, another study found that increased green level of the roadside environment, such as grass and shrubs, reduced the negative psychological state of the drivers [38].

(Line 129-130) In particular, PRS estimated the restorative value of visual environments to measure five factors (being away, fascination, coherence, scope, and compatibility) [27,42].

(Line 131-132) In addition, a PRS-11 items scale consisting of 11 items was developed to assess four factors (being away, fascination, coherence, and scope) [41].

(Line 134-136) The restoration outcome scale (ROS) using six items assessed restorative experiences in one’s favorite place [44].

(Line 142-143) For instance, one study found that cities, lakes, mountains, and seas help people regain their focus by creating restorative environments [27].

(Line 147-148) Restorative environment was measured using four items: being away, fascination, coherence, and scope [41].

(Line 162-164) Life satisfaction is measured using multi-item scales, such as the Satisfaction with Life Scale (SWLS) is a short five-item instrument to measure the cognitive factors of the subjective wellbeing [53,54].

(Line 170-172) The frequency of using residential quarter green spaces (RQGS) could positively impact residents’ life satisfaction [38].

(Line 172-173) Similarly, another study showed that after greening a schoolyard located in a city, students’ stress levels decreased, and psychological well-being improved [48].

(Line 205-207) Psychological well-being was defined using six attributes: self-acceptance, positive relations with others, autonomy, environmental mastery, purpose in life, and personal growth [60].

(Line 212-214) Another study that three restorative environmental factors (being away, fascination, and compatibility) that could significantly affect psychological well-being [62].

(Line 354-356) A previous study supported this result that “being away” from monastery played a role in increasing satisfaction for first-time visitors but did not play an important role for repeat visitors [75].

(Line 376-377) In particular, participation time in nature-based activities, rather than visit periods to restorative environments, affects well-being through restorative experience [32].

Point 2: Line 81-87. The low number of study on this term is not reasonable. Authors should explain why this study should be done, from the perspective of how people can benefit from it.

Response 2: Thank you for your thoughtful comment. We have added more references about ART and explanations of benefit of this study.

(Line 80-83) Thus, if we empirically verify the effect of apartment complex landscaping space (ACLS) on people's mental health, it could contribute to providing a green space design for residents’ psychological restoration and well-being.

(Line 84-88) From this perspective, attention restoration theory (ART) is widely used to explain the psychological recovery effect in the natural environment [25,26]. Although several studies based on ART have revealed that the natural environment significantly influences psychological restoration and relaxation [25-33], few studies have addressed these relation-ships from the perspective of ACLS.

(Reference) Berman, M.G.; Jonides, J.; Kaplan, S. The cognitive benefits of interacting with nature. Psychological science 2008, 19, 1207-1212; Korpela, K.; Borodulin, K.; Neuvonen, M.; Paronen, O.; Tyrväinen, L. Analyzing the mediators between nature-based outdoor recreation and emotional well-being. Journal of environmental psychology 2014, 37, 1-7; White, M.P.; Pahl, S.; Ashbullby, K.; Herbert, S.; Depledge, M.H. Feelings of restoration from recent nature visits. Journal of Environmental Psychology 2013, 35, 40-51.

Point 3: Line 232-234. Please explain how authors “verifying the reliability and validity of the measurement items on the questionnaire through a preliminary survey targeting 50 people”. Describe the detail, what kind of method and so on.

Response 3: Thank you for your thoughtful comment. We have added more explanations about a preliminary survey in the section 3.1 (Data collection).

(Line 236-239) In particular, we verified the reliability and validity of the measurement through exploratory factor analysis (EFA). The result of the factor loadings was greater than 0.5, and Cronbach's alpha coefficient values were greater than 0.7 (Figure 2).

Reviewer 4 Report

Goodness of index such as q value, IFI ,CF,I, Rmsea, etc are not reported yet. Authors need to present it.

Author Response

Point 1: Goodness of index such as q value, IFI ,CF,I, Rmsea, etc are not reported yet. Authors need to present it.

Response 1: Thank you for your thoughtful comment. According to your sugestion, we have revised it.

(Line 301-309) In addition, the PLS model fit of can be explained by the standardized root mean square residual (SRMR) and normal fit index (NFI). The former allows evaluation of the average magnitude of discrepancy between the observed and expected correlations [73]. In general, if the SRMR value is less than 0.10, the model fit is acceptable [73]. The latter represents that difference between one the chi-square value of the proposed model divided by the chi-square value of the null model, and the closer the NFI value is to 1, the better the model fit [74]. The SRMR and NFI values of this study meet the criterion values of 0.05 and 0.84, respectively, the model fit of the study is considered acceptable.

(Reference) Hair, J. F., Henseler, J., Dijkstra, T. K., & Sarstedt, M. (2014). Common beliefs and reality about partial least squares: comments on Rönkkö and Evermann; Lohmoller, J. B. (1988). The PLS program system: Latent variables path analysis with partial least squares estimation. Multivariate Behavioral Research, 23(1), 125-127.

Reviewer 5 Report

 I checked the revised manuscript. Thank you for providing a point-by-point responses. The manuscript has been amended accordingly, as commented by reviewer.

Author Response

Point 1: I checked the revised manuscript. Thank you for providing a point-by-point responses. The manuscript has been amended accordingly, as commented by reviewer.

Response 1: We greatly appreciate your insightful and helpful comments on our manuscript.
